# Strong metal-support interaction promoted scalable production of thermally stable single-atom catalysts

Kaipeng Liu [1,2,9], Xintian Zhao[3,9], Guoqing Ren[1,2], Tao Yang[3], Yujing Ren[1,2], Adam Fraser Lee [4], Yang Su[1], Xiaoli Pan[1], Jingcai Zhang[1], Zhiqiang Chen[1], Jingyi Yang[1,2], Xiaoyan Liu[1], Tong Zhou[5], Wei Xi[5], Jun Luo [5], Chaobin Zeng[6], Hiroaki Matsumoto[6], Wei Liu[7], Qike Jiang[7], Karen Wilson[4], Aiqin Wang[1,7], Botao Qiao [1,8✉], Weizhen Li [1✉] & Tao Zhang[1,2,7✉]

Single-atom catalysts (SACs) have demonstrated superior catalytic performance in numerous heterogeneous reactions. However, producing thermally stable SACs, especially in a simple and scalable way, remains a formidable challenge. Here, we report the synthesis of Ru SACs from commercial $RuO_2$ powders by physical mixing of sub-micron $RuO_2$ aggregates with a $MgAl_{1.2}Fe_{0.8}O_4$ spinel. Atomically dispersed Ru is confirmed by aberration-corrected scanning transmission electron microscopy and X-ray absorption spectroscopy. Detailed studies reveal that the dispersion process does not arise from a gas atom trapping mechanism, but rather from anti-Ostwald ripening promoted by a strong covalent metal-support interaction. This synthetic strategy is simple and amenable to the large-scale manufacture of thermally stable SACs for industrial applications.

[1] CAS Key Laboratory of Science and Technology on Applied Catalysis, Dalian Institute of Chemical Physics, Chinese Academy of Sciences, 116023 Dalian, China. [2] University of Chinese Academy of Sciences, 100049 Beijing, China. [3] School of Science, MOE Key Laboratory for Non-Equilibrium Synthesis and Modulation of Condensed Matter, Xi'an Jiaotong University, 710049 Xi'an, China. [4] Applied Chemistry & Environmental Science, RMIT University, Melbourne, VIC 3000, Australia. [5] Center for Electron Microscopy and Tianjin Key Lab of Advanced Functional Porous Materials, Institute for New Energy Materials and Low-Carbon Technologies, School of Materials Science and Engineering, Tianjin University of Technology, 300384 Tianjin, China. [6] Hitachi High-Technologies (Shanghai) Co., Ltd, 201203 Shanghai, China. [7] State Key Laboratory of Catalysis, Dalian Institute of Chemical Physics, Chinese Academy of Sciences, 116023 Dalian, China. [8] Dalian National Laboratory for Clean Energy, 116023 Dalian, China. [9] These authors contributed equally: Kaipeng Liu, Xintian Zhao. ✉email: bqiao@dicp.ac.cn; weizhenli@dicp.ac.cn; taozhang@dicp.ac.cn

In recent years, single-atom catalysts (SACs) have attracted considerable attention as a means by which to maximize precious metal utilization and generate well-defined, uniform active sites[1–9]. SACs exhibit superior catalytic performance (activity and/or selectivity) for thermal oxidation[1,10–12] and hydrogenation[9,13–17], electrochemistry[18–23], and industrially important processes such as the water–gas shift reaction, C-C coupling, C-H activation, and methanol reforming[11,24–27]. Counter-intuitively, SACs were recently reported to exhibit better stability than their nanoparticle (NP) counterparts, highlighting their potential for commercial applications[28,29].

Various strategies have been developed for the fabrication of SACs. Atomic layer deposition and mass-selected soft-landing methods offer precise and controllable synthesis of well-designed SACs[30–32]; however, their scale-up is hindered by high production costs and low catalyst yields[33,34]. Wet chemical routes, such as incipient wetness impregnation (IWI) and strong electrostatic adsorption methods, are common in laboratory-scale catalyst synthesis. However, they are best suited to low metal loadings[1,35,36] and are often time-consuming and process intensive, which is unfavorable for scale-up[18,37]. In addition, the thermal stability of the resulting SACs is typically poor[18,35]. Large-scale synthesis of thermally stable SACs therefore remains problematic.

Atom trapping is an effective method to produce thermally stable SACs[38–40] but still relies on wet chemistry to prepare the nanocatalysts as precursors. Based on atom trapping, Wu and Li have developed several approaches, including thermal emitting and solid diffusion, to transform bulk metals into single atoms[18,41,42] and hence open a pathway to scalable SAC production. Unfortunately, these approaches are mainly limited to carbon or N-doped carbon supports and require ammonia or HCl, which present environmental challenges.

Herein, we report a simple route to prepare thermally stable Ru SACs directly from commercial $RuO_2$ powders by heating of physical mixtures of $RuO_2$ and strongly interacting supports. Transformation of $RuO_2$ powders into isolated Ru atoms is promoted by a strong covalent metal–support interaction (CMSI) with $MgAl_{1.2}Fe_{0.8}O_4$. The resulting Ru SAC has excellent thermal stability and improved activity for $N_2O$ decomposition at low and high concentrations. This simple and low-cost synthesis paves a way for the large-scale production of thermally stable SACs with high metal loadings for industrial applications.

## Results

### Synthesis and structure of Ru SACs.

We recently observed that Pt NPs supported on iron oxides can be dispersed into single atoms upon high-temperature calcination[43]. It transpires that a strong CMSI between Fe and Pt is critical to the dispersion process, which also occurs for Fe-doped (but not undoped) $Al_2O_3$. The chemical similarity of Pt group metals suggests that such interaction may provide a general approach to fabricate thermally stable SACs[29]. Spinels, mixed metal oxides with well-defined structures and excellent thermal stability, are ideal supports for the fabrication of thermally stable catalysts[44,45]. The synthesis of a Ru SAC from a Fe-substituted $MgAl_2O_4$ spinel was therefore explored to verify the generality of this strategy.

A $MgAl_2O_4$ spinel (designated as MA) and Fe-substituted $MgAl_2O_4$ spinel ($MgAl_{1.2}Fe_{0.8}O_4$, designated as MAFO) were prepared by solvothermal synthesis and subsequent 700 °C calcination for 5 h as described in the "Methods" section. Supported Ru/MAFO analogs were prepared by conventional IWI of ruthenium(III) acetylacetonate and subsequent calcination at 500 °C (Ru/MAFO-IWI-500) or 900 °C (Ru/MAFO-IWI-900).

X-ray diffraction (XRD) patterns showed that MAFO comprised a pure crystalline spinel phase (Supplementary Fig. 1), indicating that Fe was uniformly incorporated throughout support. The MAFO surface area was far higher than that of commercial $Fe_2O_3$[43] (~100 vs. <10 $m^2 g^{-1}$, respectively, Supplementary Table 1) offering the prospect of a higher density of anchor sites to immobilize metal atoms. High-angle annular dark-field scanning transmission electron microscopy (HAADF-STEM) revealed small Ru NPs in the uncalcined IWI sample (Supplementary Fig. 2), which disappeared after 900 °C calcination (Supplementary Fig. 3a–c) implying their dispersion into single atoms[43]. Aberration-corrected (AC) HAADF-STEM images confirmed the formation of uniformly dispersed Ru single atoms (Supplementary Fig. 3d–f). In contrast, lower-temperature (500 °C) calcination resulted in severe sintering of impregnated Ru species into sub-micron $RuO_2$ aggregates (Supplementary Fig. 4), consistent with our observations for Pt sintering over $Fe_2O_3$ following low-temperature calcination[43]. Since the Ru/MAFO-IWI-900 sample transitions through lower temperatures during the heating process, we reasoned that these large $RuO_2$ aggregates must be thermodynamically unstable and hence should be susceptible to re-dispersion when subject to a further high-temperature calcination. HAADF-STEM confirmed that 900 °C calcination of the Ru/MAFO-IWI-500 sample resulted in complete loss of the $RuO_2$ aggregates (Supplementary Fig. 5).

The remarkable efficacy of MAFO for dispersing sub-micron Ru aggregates into single atoms at high temperatures inspired us to explore whether commercial $RuO_2$ powders (rather than costly organometallic complexes) could be used as the metal precursor to synthesize Ru SACs. To maximize the interface between commercial $RuO_2$ powders (containing sub-micron particles) and MAFO, a physical mixture of the two components was simply ground and calcined at either 900 °C for 5 h in air (denoted as $Ru_1$/MAFO-900) or 500 °C (denoted as Ru/MAFO-500). This synthesis is illustrated in Supplementary Fig. 6; the nominal Ru loadings in both cases were 2 wt%.

The resulting Ru/MAFO-500 sample contained sub-micron $RuO_2$ aggregates (Fig. 1a–c and Supplementary Fig. 7, which are insoluble in aqua regia solution, Supplementary Fig. 8) of similar size to the parent $RuO_2$ powders (Supplementary Fig. 9) consistent with Ru/MAFO-IWI-500. However, no Ru NPs or nanoclusters (NCs) were apparent by low-magnification HAADF-STEM for the $Ru_1$/MAFO-900 sample (Supplementary Fig. 10a–c), despite element analysis confirming the presence of 2 wt% Ru (Supplementary Fig. 10d and Supplementary Table 2). The absence of Ru aggregates in $Ru_1$/MAFO-900 must therefore reflect dispersion, not loss, of Ru species; indeed AC-HAADF-STEM evidenced a high density of uniformly dispersed Ru single atoms on the MAFO spinel support (Fig. 1e, f and Supplementary Fig. 10e–g).

XRD corroborated the preceding observations (Fig. 2a). The untreated physical mixture exhibits reflections characteristic of the rutile structure of $RuO_2$ and the MAFO support; the former remain visible following 500 °C calcination but are completely lost after 900 °C consistent with Ru dispersion. MAFO reflections are slightly sharpened by the 900 °C calcination, indicating partial support sintering in accordance with the concomitant decrease in Brunauer–Emmett–Teller (BET) surface area (Supplementary Table 1). The chemical state of Ru was investigated by X-ray photoelectron spectroscopy (XPS). Note that the C $1s$ and Ru $3d$ photoemissions overlap, and hence Ru $3p$ XP spectra were measured, revealing identical Ru $3p_{3/2}$ binding energies of 463.2 eV for the $Ru_1$/MAFO-900 and Ru/MAFO-500 samples (Fig. 2b), characteristic of $Ru^{4+}$ species[46,47]. However, the spectrum intensity for $Ru_1$/MAFO-900 is significantly higher than that for Ru/MAFO-500, in good agreement with its much higher dispersion. X-ray absorption spectroscopy was also measured to elucidate the local chemical environment of Ru within both

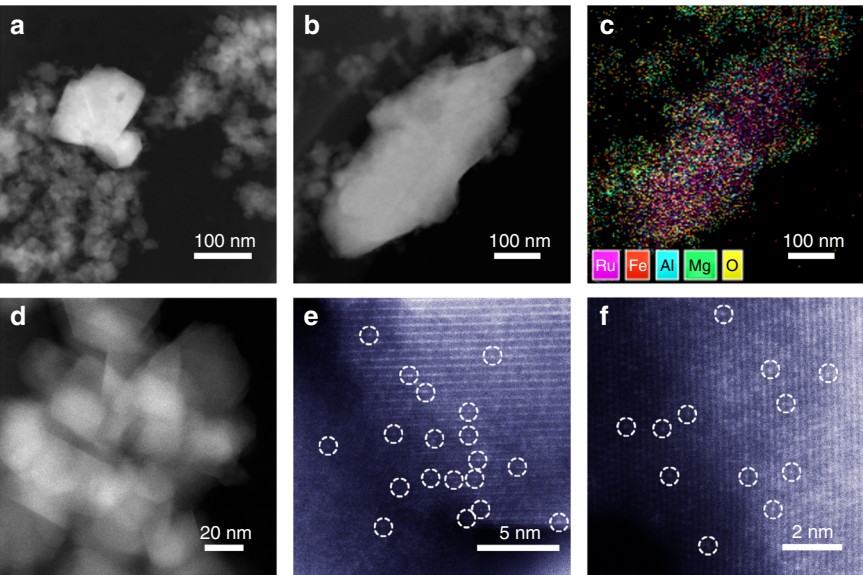

**Fig. 1 HAADF-STEM characterization of Ru/MAFO samples. a**, **b** HAADF-STEM images of Ru/MAFO-500 sample. **c** Energy dispersive X-ray spectroscopy elemental mapping results of Ru/MAFO-500 sample. **d**–**f** AC-HAADF-STEM images of $Ru_1$/MAFO-900 sample.

samples (Fig. 2c). The absorption edge energies of $Ru_1$/MAFO-900 and Ru/MAFO-500 were identical and matched that for $RuO_2$, consistent with the presence of $Ru^{4+}$ species observed by XPS; however, the X-ray absorption near-edge structure (XANES) of $Ru_1$/MAFO-900 differed from that of Ru/MAFO-500 and $RuO_2$, i.e., Ru atoms in $Ru_1$/MAFO-900 are in a different coordination environment to those in $RuO_2$ NPs/aggregates[48,49]. Fourier transforms of the corresponding extended X-ray absorption fine structure (EXAFS) reveal two well-defined coordination shells at ~1.97 and 3.54 Å for $RuO_2$ and Ru/MAFO-500 associated with Ru-O and Ru-O-Ru scattering contributions, respectively (Fig. 2d, Supplementary Fig. 11, and Supplementary Table 3)[50,51]. In contrast, no Ru-O-Ru or Ru-Ru contributions were observed for $Ru_1$/MAFO-900, akin to EXAFS data for atomically dispersed Pd over mesoporous $Al_2O_3$[52] and Pt over $Fe_2O_3$[43], unambiguously evidencing Ru single atoms. In addition to a nearest neighbor Ru-O shell, significant Ru-Fe scattering was observed for $Ru_1$/MAFO-900 consistent with a strong chemical bonding to $FeO_x$ surface sites[43]. We can therefore conclude that high-temperature calcination of a physical mixture of commercial $RuO_2$ powders and MAFO results in a 2 wt% Ru SAC.

Catalysts with higher Ru loadings such as 2.5 and 3 wt% were further prepared with the same procedure. Obvious $RuO_2$ diffraction peaks were observed for both samples (Supplementary Fig. 12), indicating that the maximum Ru loading is in fact around 2 wt%. We estimated the theoretical maximum loading of dispersed Ru atoms over MAFO support by assuming only surface Fe as the stabilization sites to be around 1.6 wt% (for details, see "Methods")[43], which agree well with the experimental data. The good consistency suggested that the Ru atoms mainly located on the surface/subsurface rather than diffused into the bulk of the support because the latter case will give rise to a much higher maximum Ru loading. The Fe content in MAFO is tunable. We further investigated the effect of Fe content by preparing three $MgAl_{2-x}Fe_xO_4$ supports with different Fe contents ($x = 0.5, 1, 1.5$). As shown in Supplementary Table 1, the substitution of Fe weakens the sintering resistance of the $MgAl_2O_4$ spinel, thus inducing a surface area decrease after being calcined at high temperatures. Meanwhile, excess Fe substitution would result in the appearance of an impure phase of iron oxide (Supplementary Fig. 13a). We then tried to synthesize Ru SACs by using the newly

synthesized pure phase materials ($MgAl_{1.5}Fe_{0.5}O_4$ and $MgAl_1Fe_1O_4$) as supports by the same procedure. The formation of $2Ru/MgAl_1Fe_1O_4$-900 SAC was confirmed by XRD and AC-HAADF-STEM characterizations (Supplementary Figs. 13b and 14). However, weak diffraction peaks of $RuO_2$ were observed in the $2Ru/MgAl_{1.5}Fe_{0.5}O_4$-900 sample, suggesting that $RuO_2$ cannot be completely dispersed on this sample. This likely reflects the low Fe content in the $MgAl_{1.5}Fe_{0.5}O_4$ spinel that cannot provide sufficient sites to stabilize all Ru single atoms, consistent with the calculated theoretical maximum Ru loading for $MgAl_{1.5}Fe_{0.5}O_4$ support (up to 1.0 wt%; for details, see "Methods"). Based on the above analysis, we propose that for the catalyst with 2 wt% Ru loading the optimized Fe ratio should be around $x = 1$. For lower Ru loading, the optimized Fe content needs further study; we believe that provided sufficient stabilizing sites are present, the smaller the Fe content the better.

**Catalytic performance of Ru/MAFO samples.** The catalytic performance of the preceding Ru/MAFO catalysts was subsequently studied for nitrous oxide ($N_2O$) decomposition, an important reaction in an environmental context and satellite propulsion systems. $N_2O$ is a potent greenhouse gas facilitating ozone depletion even at very low concentrations[53–55]. However, at high concentrations, $N_2O$ is a potential "green" propellant in the aerospace sector[56–58]. Catalytic decomposition of $N_2O$ into $N_2$ and $O_2$ is therefore a promising route to eliminate (undesirable) low concentrations in the atmosphere and exploit high concentrations as a fuel, and hence both limits (1000 ppm and 20 vol% $N_2O$ in Ar) were explored in this work (Fig. 3a). The $Ru_1$/MAFO-900 SAC exhibited much greater activity than Ru/MAFO-500 at both $N_2O$ concentrations, reflected in lower light-off temperatures. $Ru_1$/MAFO-900 also displayed excellent stability at 550 °C for decomposition of low $N_2O$ concentration, with conversion remaining ~76% for 100 h on-stream (Fig. 3b); although Ru/MAFO-500 was also very stable under these conditions, $N_2O$ conversion was only ~25% (a small activity increase at long reaction times may reflect dispersion of small amount of the sub-micron $RuO_2$ aggregates). XRD (Supplementary Fig. 15) and HAADF-STEM (Supplementary Fig. 16) evidenced no Ru NCs or NPs for $Ru_1$/MAFO-900 post-reaction, demonstrating

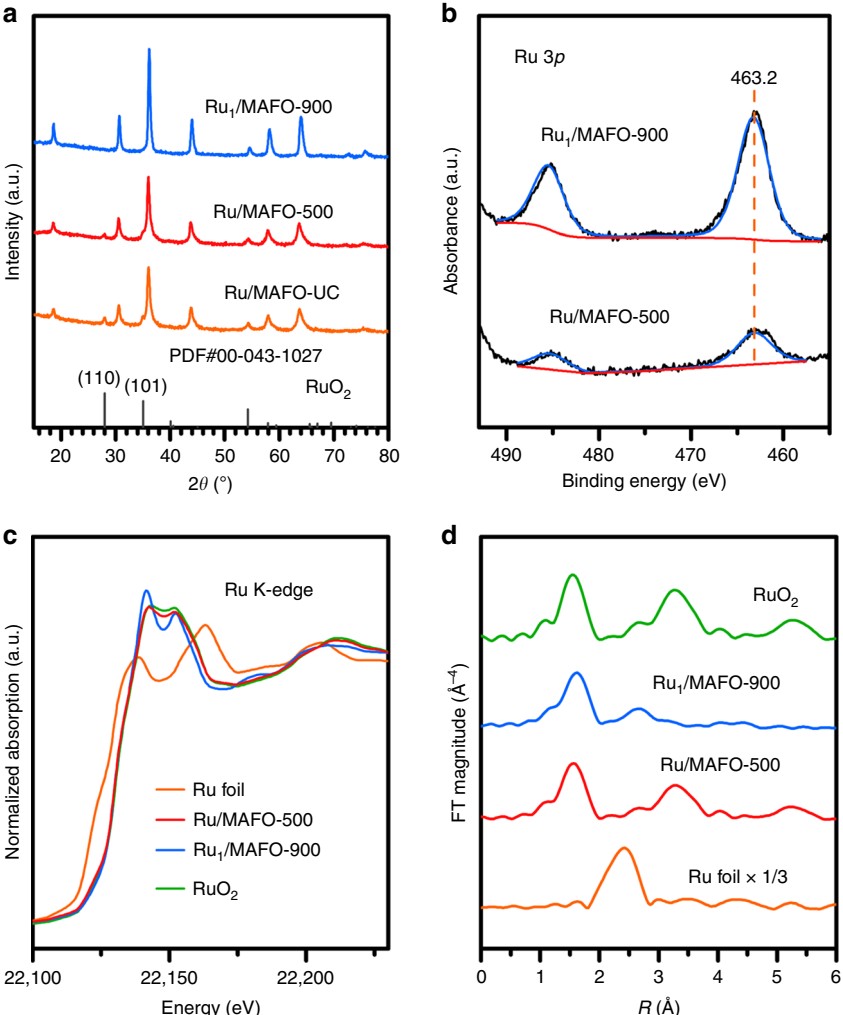

**Fig. 2 Structural characterizations of Ru/MAFO samples. a** XRD patterns of Ru/MAFO samples and reference materials (PDF#00-043-1027 is the JCPDS card number of $RuO_2$). **b** Ru $3p$ XPS of Ru/MAFO samples. **c** Normalized Ru $K$-edge XANES of Ru/MAFO samples and references. **d** Fourier transforms of $k^3$-weighted Ru $K$-edge EXAFS spectra of Ru/MAFO samples and references (without phase correction).

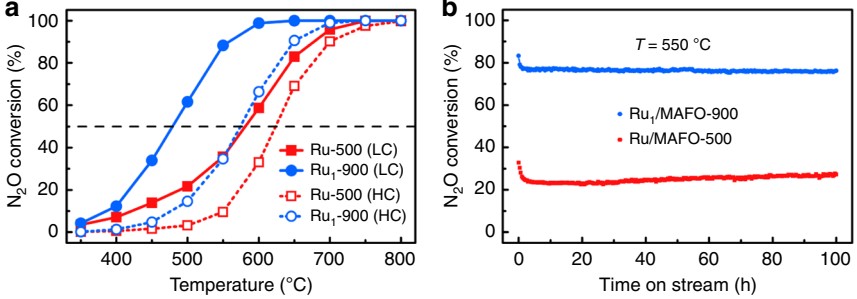

**Fig. 3 Catalytic performance of Ru/MAFO samples for $N_2O$ decomposition. a** $N_2O$ conversion as a function of reaction temperature on Ru/MAFO samples at low (1000 ppm $N_2O$, solid symbol) and high (20 vol% $N_2O$, open symbol) concentrations. Reaction conditions: 100 mg catalyst; gas flow, 33.3 mL min$^{-1}$; GHSV = 20,000 mL g$_{cat}$$^{-1}$ h$^{-1}$; Ar balance. **b** $N_2O$ conversion as a function of reaction time on Ru/MAFO samples in low-concentration $N_2O$ decomposition tested at 550 °C. Reaction conditions: 100 mg catalyst; gas flow, 33.3 mL min$^{-1}$; GHSV = 20,000 mL g$_{cat}$$^{-1}$ h$^{-1}$.

the Ru single atoms are extremely stable under our reaction conditions; elemental analysis also showed no loss of Ru (Supplementary Table 2). Interestingly, decomposition of high $N_2O$ concentration at elevated temperatures (800 °C) over Ru/MAFO-500 resulted in a step change in conversion after only a few minutes on-stream (Supplementary Fig. 17), which we attribute

to dispersion of the initial $RuO_2$ aggregates; a similar phenomenon was observed for $CH_4$ oxidation over Pt/$Fe_2O_3$[43]. XRD and AC-HAADF-STEM characterization of the post-reaction sample (Supplementary Figs. 18 and 19) support this proposal and demonstrate that atomically dispersed Ru is the main active site for high-temperature $N_2O$ decomposition.

**Mechanism of RuO₂ dispersion**. RuO₂ powders/aggregates can be dispersed into single atoms on MAFO by high-temperature calcination. We believe that substituted Fe plays a critical role in trapping and stabilizing Ru atoms or RuO₂ single clusters through a CMSI effect[43], a conjecture easily verified by control experiments with an Fe-free spinel (MA). As anticipated, XRD indicated that RuO₂ aggregates are not dispersed into isolated atoms over the MA support by high-temperature calcination (Supplementary Fig. 20) but rather undergo sintering resulting in sharper RuO₂ reflections. AC-HAADF-STEM confirmed that large RuO₂ aggregates were retained in the Ru/MA-900 sample, although a small number of RuO₂ NPs or NCs were also observed (Supplementary Fig. 21).

The question arises as to the mechanism of Ru dispersion. Gas-phase atom trapping is a common process by which high-temperature dispersion may occur but is usually accompanied by metal losses[18,40]. In the present case, no detectable Ru loss was observed, suggesting the operation of a different mechanism, and confirmed by the following control experiments. High-temperature calcination of RuO₂ and MAFO was repeated using different locations for the two components (Supplementary Fig. 22): RuO₂ powders were placed (a) on the surface of or (b) beneath the MAFO spinel or (c) randomly mixed with the spinel by applying a mechanical vibration. Considering that RuO₂ can oxidize to form volatile RuO₃ and/or RuO₄ at very high temperatures[59–61], if gas-phase atom trapping dominated the dispersion process, then all three geometries should result in efficient Ru dispersion over MAFO since volatilized gas-phase atoms can diffuse to large (in cm level) distances[18,61]. In practice, the RuO₂ powders were unchanged and clearly visible as a separate phase following calcination in scenarios (a) and (b) (Supplementary Fig. 22), and we can therefore discount a gas-phase atom trapping mechanism. This is in accordance with additional control experiments in which RuO₂ powders were calcined without the MAFO support, which resulted in minimal weight loss (<10%) under static or flowing conditions (Supplementary Table 4, entry 1, 2). Note that in scenario (c), although the ochre color of the calcined sample darkened somewhat (a characteristic of Ru₁/MAFO-900, Supplementary Fig. 23), XRD reflections of RuO₂ remained visible (Supplementary Fig. 24), and black insoluble substances were observed following dissolution of

the MAFO support in aqua regia (Supplementary Fig. 25) consistent with large RuO₂ aggregates. The Ru loading in the vibration mixed Ru/MAFO-VM-900 sample was only 0.72 wt% (Supplementary Table 2), far less than the nominal loading, indicating that only a small amount of RuO₂ aggregates were dispersed over the spinel. We can therefore conclude that intimate physical mixing (PM) of RuO₂ and MAFO prior to their calcination is essential to maximize the resulting dispersion of Ru single atoms.

The control experiment highlighted that RuO₂ volatilization was minimized under an inert environment (Supplementary Table 4, entry 3), and hence RuO₂ dispersion over MAFO was also attempted by annealing at 900 °C under Ar and He atmospheres (conditions strongly disfavoring gas-phase atom trapping). In both cases, XRD confirmed the loss of RuO₂ reflections following 5 h anneals (Supplementary Fig. 26) consistent with at least partial Ru dispersion. The resulting Ru/MAFO loadings of ~1.6 wt% (Supplementary Table 2) were slightly lower than the nominal 2 wt% value, suggesting that a small proportion of the parent RuO₂ remained intact, and indeed a subsequent aqua regia treatment of both Ru/MAFO materials revealed trace insoluble component (Supplementary Fig. 27). Extended annealing under He increased the final Ru loading to 2 wt% (Supplementary Table 2), indicating complete dispersion of this residual RuO₂ into single atoms over the MAFO support. In summary, there is no evidence that Ru volatilization and subsequent gas-phase atom trapping is mainly responsible for RuO₂ dispersion.

The kinetics of RuO₂ dispersion by air calcination was also explored. XRD showed the immediate disappearance of RuO₂ reflections on heating to 900 °C (0-h sample, Supplementary Fig. 28), although HAADF-STEM highlighted trace residual RuO₂ aggregates that required ≥1 h at 900 °C to fully disperse into Ru single atoms (Supplementary Figs. 29 and 30). The dispersion process was directly visualized by in situ AC-HAADF-STEM and simultaneous secondary electron (SE) detection (Fig. 4, Supplementary Fig. 31, and Supplementary Movie 1); a large RuO₂ aggregate in the initial RuO₂+MAFO physical mixture was randomly selected and tracked in real time during calcination. The size and morphology of the RuO₂ aggregate were unchanged at <900 °C, at which point a melting-like phenomenon

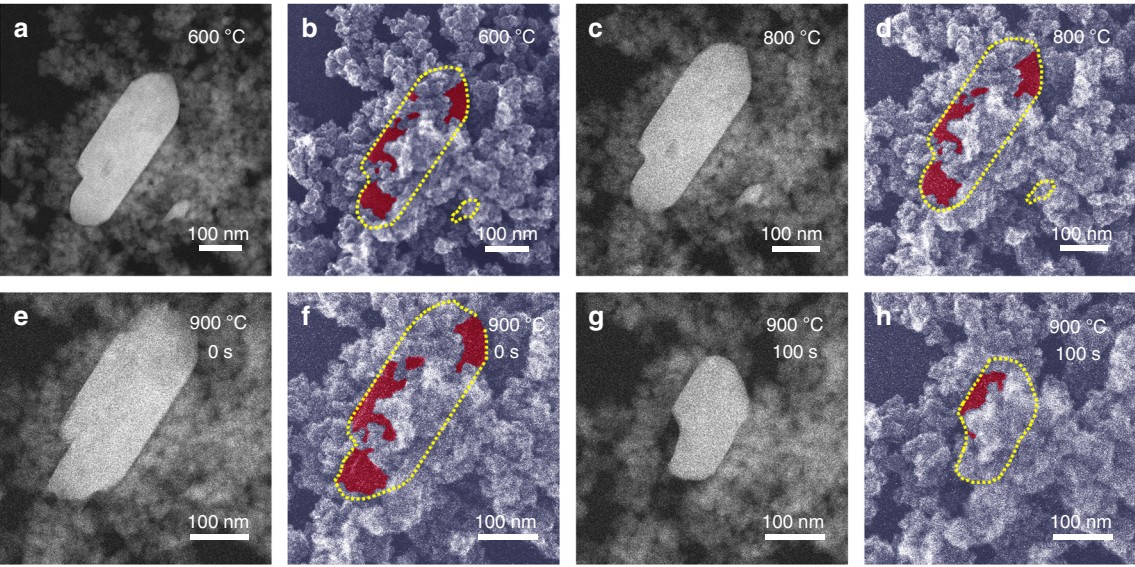

**Fig. 4 In situ characterization of RuO₂ dispersion. a**, **c**, **e**, **g** In situ AC-HAADF-STEM images and **b**, **d**, **f**, **h** corresponding SE images of a RuO₂+MAFO physical mixture after calcination at 600, 800, and 900 °C (0 s, 100 s) under flowing O₂ (2 mL min⁻¹ and 3.5 Pa). Yellow dashed lines in the SE images silhouette the RuO₂ aggregate, and red regions indicate exposed RuO₂ surfaces.

commenced, coincident with achieving the Tammann temperature of $RuO_2$[62]. Over the following 100 s at 900 °C, the $RuO_2$ aggregate shrank by approximately 50% in all dimensions. Note that SE imaging revealed that the $RuO_2$ aggregate was partially embedded in the granular MAFO support and did not move during heating (this helps guide our proposed dispersion mechanism below). Another smaller (~100 × 100 nm) $RuO_2$ aggregate underwent similar melting and shrinking, being fully dispersed after 30 min at 900 °C (Supplementary Fig. 32).

In situ electron microscopy cannot (yet) directly record the movement of individual atoms over practical catalysts under such conditions; however, the preceding images enable us to exclude certain dispersion processes such as Brownian motion of $RuO_2$ aggregates throughout the MAFO matrix, distributing Ru atoms/ $RuO_2$ sub-units as it passes. The only plausible dispersion model is therefore an anti-Ostwald ripening process wherein Ru atoms/ $RuO_2$ sub-units break away from static $RuO_2$ aggregates and diffuse across the MAFO surface until being trapped by a CMSI. The rapidity of $RuO_2$ dispersion over MAFO vs. MA supports at 900 °C suggests that CMSI involving $FeO_x$ sites may promote such ripening.

To further verify the CMSI between $RuO_2$ and $FeO_x$, we performed a $H_2$ temperature-programmed reduction ($H_2$-TPR) characterization. As shown in Supplementary Fig. 33, on Ru/MA-500, Ru/MA-900, and Ru/MAFO-500 samples two reduction peaks were observed between 100 and 200 °C. The former corresponds to the reduction of $RuO_2$ to RuO while the latter is ascribed to the reduction of RuO to Ru metal[63,64]. The slightly higher temperature for the reduction of RuO on Ru/MAFO-500 than that on Ru/MA-500 may suggest that Ru species interact stronger with MAFO than with MA. Of more importance, the low-temperature reduction of Ru nearly vanished on the $Ru_1$/ MAFO-900 sample with only a very tiny reduction peak (marked by arrow). The majority of the Ru species must have been reduced together with Fe at higher temperatures, suggesting a strengthened interaction between $RuO_2$ and $FeO_x$ after being calcined at 900 °C. A quantitative analysis (Supplementary Table 5) revealed that the $H_2$ consumptions on Ru/MA-500, Ru/MA-900, and Ru/ MAFO-500 samples are similar to the theoretical one for complete reduction of $RuO_2$ to Ru. However, for $Ru_1$/MAFO-900 sample, the $H_2$ consumption of the tiny reduction peak is only about 1/27 of the theoretical one, corresponding to a reduction of Ru loading of ~0.07 wt%. We propose that these Ru species may be stabilized by Mg or Al sites since the MA support itself can stabilize very low loading of Ru single atoms[65].

A recent theoretical study proposed that strong metal atom–support interactions can decrease the activation energy (and hence promote the occurrence) of Ostwald ripening[66], in good agreement with our experimental observations. The possibility that a CMSI promotes $RuO_2$ dispersion in our system was investigated by density functional theory (DFT) calculations (for details, see "Methods"). The small $RuO_2$ clusters ($Ru_5O_{10}$ or $Ru_{10}O_{20}$) supported on $MgAl_2O_4(100)$ and two-layer Fe-substituted $MgAl_2O_4(100)$, respectively, were studied for comparison. Geometry optimization revealed that either the longest or the average Ru–Ru distance in $RuO_2$ clusters supported on Fe-substituted $MgAl_2O_4(100)$ were significantly elongated, compared to those on $MgAl_2O_4(100)$ surface (Supplementary Fig. 34 and Supplementary Table 6). In particular, one $RuO_2$ moiety evidently moves away from the $RuO_2$ cluster on Fe-substituted $MgAl_2O_4(100)$. Further calculations on the binding energy and reaction Gibbs free energy showed that the farthest $RuO_2$ moiety dissociation from the $RuO_2$ cluster supported on Fe-substituted $MgAl_2O_4(100)$ surface is preferred over on $MgAl_2O_4(100)$ surface (Supplementary Table 7), which comes from the Fe effect on the metal–support interaction, as confirmed by electron density

difference maps in Supplementary Fig. 35. These results revealed that the presence of Fe atom weakens Ru–Ru interaction in cluster and promotes $RuO_2$ dispersion. We therefore propose that a strong CMSI between $FeO_x$ sites in the MAFO support and $RuO_2$ aggregates promotes anti-Ostwald ripening of Ru atoms/ $RuO_2$ sub-units.

**Scalable production of SACs.** Scale-up represents a key barrier to progressing SACs from intellectual curiosity to practical solution for industrial chemical processes. The utility of our simple mixing/calcination protocol was therefore exploited to prepare 10 g $Ru_1$/MAFO ($Ru_1$/MAFO-10g-900, Supplementary Fig. 36). Structural characterization by XRD, HAADF-STEM, XAFS, and elemental analysis unambiguously demonstrated a successful scale-up (Supplementary Figs. 37–39 and Supplementary Tables 2 and 3). Since Fe is critical to the dispersion of $RuO_2$ aggregates into single atoms over the MAFO spinel, we subsequently explored whether commercial $Fe_2O_3$ alone could provide a suitable support for generating a Ru SAC on a kilogram scale. Corresponding XRD, HAADF-STEM, and elemental analysis confirmed that almost all the $RuO_2$ was dispersed into single atoms over the $Fe_2O_3$ support following a 900 °C calcination (Supplementary Figs. 40 and 41 and Supplementary Table 2). Note that the surface area of commercially available $Fe_2O_3$ is only ~4 $m^2 g^{-1}$ and the theoretical maximum loading of Ru for SACs is ~0.4 wt%. Thus we used a low Ru loading of 0.3 wt% to ensure that the number of Ru atoms are smaller than the stabilization sites. Although the $Ru_1/Fe_2O_3$-1000g-900 sample shows much lower activity compared with the $Ru_1$/MAFO-900 sample (Supplementary Fig. 42) due probably to the lower redox activity and/ or the significantly lower surface area of $Fe_2O_3$, the ability to prepare 1 kg of SAC by mixing and heating two commercial bulk oxides may have a profound influence on the future direction of catalyst manufacturing.

## Discussion
We have developed a simple strategy to prepare Ru SACs by PM of commercially available $RuO_2$ powders with Fe-containing supports. $RuO_2$ powders undergo complete dispersion into isolated single atoms following high-temperature treatment under oxidizing and inert atmospheres. A strong metal–support interaction between Ru and Fe plays a critical role not only in trapping and stabilizing Ru atoms but also in promoting the ripening of $RuO_2$ aggregates. The approach is simple, general, environmentally friendly, and highly scalable, unlocking the large-scale manufacture of thermally stable SACs for industrial applications.

## Methods
**Chemical.** Magnesium nitrate hexahydrate (≥99%, Damao Chemical Reagent), aluminum isopropoxide (≥98%, Aladdin), iron(III) acetylacetonate (≥98%, Aladdin), ethanol (Sinopharm Chemical Reagent), magnesium acetate tetrahydrate (≥99%, Damao Chemical Reagent), ruthenium(III) acetylacetonate (97%, Aladdin), toluene (Sinopharm Chemical Reagent), ruthenium(IV) oxide ($RuO_2$, 99.9%, Aladdin), iron(III) oxide ($Fe_2O_3$, ≥99%, Damao Chemical Reagent), hydrochloric acid (Damao Chemical Reagent), nitric acid (Damao Chemical Reagent), and quartz sand (Damao Chemical Reagent) were used without any further purification.

**Preparation of MA spinel.** $MgAl_2O_4$ spinel (designated as MA) was prepared by hydrolysis of aluminum isopropoxide and magnesium acetate tetrahydrate in ethanol. In all, 0.15 molar of magnesium acetate tetrahydrate and 0.30 molar of aluminum isopropoxide were mixed in 900 mL of ethanol and sealed in a 2-L autoclave. The mixture was heated to 120 °C and held there for 10 h, then increased to 160 °C and held there for another 10 h under vigorous stirring. After cooling to room temperature, the obtained product was filtrated and then dried at 120 °C for 1 h, and finally calcined in ambient air at 700 °C for 5 h with a heating rate of 2 °C $min^{-1}$, resulting in the formation of MA spinel with pure spinel crystal phase.

**Preparation of MAFO spinels**. $MgAl_{1.2}Fe_{0.8}O_4$ spinel (designated as MAFO) was prepared by hydrolysis of aluminum isopropoxide and iron(III) acetylacetonate with magnesium nitrate hexahydrate in ethanol. In all, 0.15 molar of magnesium nitrate hexahydrate, 0.18 molar of aluminum isopropoxide, and 0.12 molar of iron (III) acetylacetonate were mixed in 900 mL of ethanol and sealed in a 2-L autoclave. The mixture was heated to 120 °C and held there for 10 h, then increased to 160 °C and held there for another 10 h under vigorous stirring. After cooling to room temperature, the obtained product was filtrated and then dried at 120 °C for 1 h, and finally calcined in ambient air at 700 °C for 5 h with a heating rate of 2 °C min$^{-1}$, resulting in the formation of MAFO spinel with pure spinel crystal phase. $MgAl_{1.5}Fe_{0.5}O_4$, $MgAl_1Fe_1O_4$, and $MgAl_{0.5}Fe_{1.5}O_4$ spinels were prepared via adjusting the ratio of aluminum isopropoxide and iron(III) acetylacetonate and used the same preparation procedure of MAFO spinel.

**Preparation of Ru/MAFO-IWI samples**. The Ru/MAFO-IWI samples (nominal weight loadings of Ru were 1 wt%) were prepared using the IWI method. The sample was synthesized using a solution of ruthenium(III) acetylacetonate in toluene. After impregnation, the sample was dried at room temperature for 24 h and 60 °C for 10 h. Then the sample was calcined in ambient air at 500/900 °C for 5 h with a heating rate of 2 °C min$^{-1}$. The resulting samples are designated as Ru/ MAFO-IWI-500 and Ru/MAFO-IWI-900. The Ru/MAFO-IWI-500 sample was further calcined in ambient air at 900 °C for 5 h with a heating rate of 2 °C min$^{-1}$. The resulting sample is designated as Ru/MAFO-IWI-500-900. And the uncalcined sample is designated as Ru/MAFO-IWI-UC.

**Preparation of Ru/MAFO and Ru/MA samples**. The Ru/MAFO and Ru/MA samples (nominal weight loadings of Ru were 2 wt%) were prepared using the PM method. Typically, 2.5 g of MAFO or MA spinel was physically mixed with 0.0673 g of $RuO_2$ with extensive grind by using an agate mortar. The obtained uncalcined mixtures are denoted as Ru/MAFO-UC and Ru/MA-UC, respectively, and were further calcined in ambient air at 500/900 °C for 5 h with a heating rate of 2 °C min$^{-1}$, designated as Ru/MAFO-500, Ru$_1$/MAFO-900 and Ru/MA-500, Ru/ MA-900, respectively. $2Ru/MgAl_{1.5}Fe_{0.5}O_4$-900 and $2Ru/MgAl_1Fe_1O_4$-900 samples were also prepared to study the effect of Fe content. For comparison, Ru/MAFO-UC was annealed in an inert atmosphere (He and Ar) at 900 °C for 5 h or 24 h with a heating rate of 2 °C min$^{-1}$, and the resulting samples are designated as Ru/ MAFO-900(He/Ar, 5 h) or Ru/MAFO-900(He, 24 h). The Ru/MAFO-UC sample was also calcined in ambient air at 900 °C for different time points with a heating rate of 2 °C min$^{-1}$ to study the dispersion mechanism. The calcination time points were 0, 1, 2, 3, 4, and 5 h, and the resulting samples are designated as Ru/MAFO-900-t, $t = 0$–5 h.

**Large-scale preparation of Ru$_1$/MAFO SAC**. In all, 10.0 g of MAFO spinel was physically mixed with 0.2689 g of $RuO_2$ and calcined in ambient air at 900 °C for 5 h with a heating rate of 2 °C min$^{-1}$. The nominal weight loading of Ru was 2 wt%. The resulting sample is designated as Ru$_1$/MAFO-10g-900.

**Large-scale preparation of Ru$_1$/Fe$_2$O$_3$ SAC**. In all, 1000 g of $Fe_2O_3$ was physically mixed with 3.9620 g of $RuO_2$ and calcined in ambient air at 900 °C for 5 h with a heating rate of 2 °C min$^{-1}$. The nominal weight loading of Ru was 0.3 wt%. The resulting sample is designated as Ru$_1$/Fe$_2$O$_3$-1000g-900.

**Three control experiments with different contact manners**. $RuO_2$ powders were located on the surface of or underneath the MAFO spinel support or randomly mixed by vibration. The comparative experiments were performed and calcined at 900 °C for 5 h with a heating rate of 2 °C min$^{-1}$. Vibration mixing was carried out using a vibrating plate, and the calcined sample is designated as Ru/MAFO-VM-900. The nominal weight loadings of Ru were 2 wt%.

**Catalyst characterization**. HAADF-STEM images were obtained on a JEOL JEM-2100F operated at 200 kV. AC-HAADF-STEM images were obtained on a FEI Titan Cubed Themis G2 300 operated at 200 kV. TEM specimens were prepared by depositing a suspension of the powdered sample on a lacey carbon-coated copper grid.

The in situ AC-HAADF-STEM/SEM experiment was performed on a Hitachi field emission scanning transmission microscope HF5000 using the MEMS heating holder, and the gas flow was controlled by MFC system. The MEMS heating holder was manufactured by Hitachi High Technologies Canada. And the chips were manufactured by Norcada Inc. The Ru/MAFO-UC sample was supported on the 50-nm-thick $Si_3N_4$ membrane. And the gas was injected to the sample area by special designed gas injection nozzle. The oxygen purity used for the in situ calcination experiment was 99.999%.

XRD patterns were recorded on a PANalytical Empyrean diffractometer equipped with a Cu Kα radiation source ($\lambda = 0.15432$ nm), operating at 40 kV and 40 mA.

The BET surface area, pore volume, and average pore size were measured with a Micromeritics ASAP 2460 instrument using adsorption of $N_2$ at 77 K. All of the

samples were degassed under vacuum at 300 °C for 5 h before the adsorption measurements.

Inductively coupled plasma optical emission spectrometry was performed on an Optima 7300DV instrument (PerkinElmer Instrument Corporation). All the samples were dissolved by using aqua regia heated on a hotplate until it was clear or continuously heated for 2 h.

X-ray fluorescence spectrometry (XRF) was performed on a PANalytical Zetium instrument. The samples were pressed into tablets before XRF analyses. In order to obtain an accurate Ru content, we prepared a calibration curve: briefly, 2.5 g of MAFO spinel was physically mixed with corresponding proportion of $RuO_2$ by using an agate mortar (for details, see Supplementary Fig. 43 and Supplementary Table 8).

XPS was measured on a Thermo Fisher ESCALAB 250Xi spectrometer equipped with an Al anode (Al Kα = 1486.6 eV), operated at 15 kV and 10.8 mA. The background pressure in the analysis chamber was $<3 \times 10^{-8}$ Pa, and the operating pressure was around $7.1 \times 10^{-5}$ Pa. The survey and spectra were acquired at a pass energy of 20 eV. Energy calibration was carried out using the C 1s peak of adventitious C at 284.8 eV.

XANES and EXAFS spectra at the Ru K-edge were recorded at the BL14W1, Shanghai Synchrotron Radiation Facility, China. A Si (311) double-crystal monochromator was used for the energy selection. The energy was calibrated by Ru foil. Ru foil and $RuO_2$ were used as reference samples and measured in the transmission mode. The Ru/MAFO-500, Ru$_1$/MAFO-900, and Ru$_1$/MAFO-10g-900 samples were measured in the transmission mode. The Athena software package was used to analyze the data.

$H_2$-TPR was carried out on a Micromeritics AutoChem II 2920 apparatus. The sample (~100 mg) was placed in the U-shaped quartz reactor and heated at 300 °C in Ar for 30 min to remove the physically adsorbed water and other contaminants. After cooling the sample down to 50 °C, the gas was switched to 10 vol% $H_2$/Ar, and the sample was heated to 900 °C at a ramp rate of 10 °C min$^{-1}$ for reduction. $H_2$ consumption during sample reduction was monitored via TCD. The amount of $H_2$ consumption was calculated with the $H_2$ peak area and calibration curve of the 10 vol% $H_2$/Ar standard gas.

**Catalytic reactions**. $N_2O$ decomposition was carried out at atmospheric pressure in a fixed-bed microreactor. In all, 100 mg of catalyst diluted with 1 g of quartz sand (40–80 mesh) was loaded into a U-shaped quartz reactor. A k-type thermocouple in a thin quartz tube was inserted into the catalyst bed to measure the temperature. The feed gas containing 1000 ppm $N_2O$ and balance Ar (low concentration) or 20 vol% $N_2O$ and balance Ar (high concentration) was passed through the reactor at 33.3 mL min$^{-1}$. Long-term stability was tested by running the reactor at 550 °C for 100 h at low $N_2O$ concentration. The test of the dispersion of $RuO_2$ was using 50 mg of the Ru/MAFO-500 catalyst diluted with 1 g of quartz sand (40–80 mesh) and performed at high $N_2O$ concentration with a high gas flow (166.7 mL min$^{-1}$). The reaction temperature increased from room temperature to 800 °C with a rate of 10 °C min$^{-1}$ and then maintained at 800 °C for 10 h. The amounts of the $N_2O$ in the inlet and outlet gas compositions were analyzed using a gas chromatograph (Echrom A91) equipped with Parapak Q packed column and a thermal conductivity detector using He as the carrier gas. For the Ru$_1$/Fe$_2$O$_3$-1000g-900 catalyst, 670 mg of catalyst diluted with 1 g of quartz sand (40–80 mesh) was loaded into a U-shaped quartz reactor in low-concentration $N_2O$ decomposition reaction under the premise of using same Ru amount.

**Computational methods**. All DFT calculations were performed with Vienna Ab-initio Simulation Package (VASP)[67,68], and the exchange-correlation energy was expressed by generalized gradient approximation of Perdew–Burke–Ernzerhof functional[69]. The projector-augmented wave method[70] was used to describe the interaction between electrons and ions. The plane-wave basis energy cutoff was set to 520 eV with the gamma point only for the Brillouin zone. The convergence criteria for the electronic structure and geometry optimization were $1 \times 10^{-4}$ eV and 0.02 eV Å$^{-1}$, respectively. Because of the strongly correlated d electrons, DFT+U calculations with corresponding $U$–$J$ values of 2.5 eV (Fe) and 2.0 eV (Ru) were employed[71,72].

**Computational models**. The $2 \times 2$ supercell model of $MgAl_2O_4(100)$[73] consists of four Al-O layers and three Mg layers, of which bottom two layers were fixed in the relaxation calculations. A 15 Å vacuum layer was added to avoid interaction between periodic structures. To model the MAFO, Al in top layers of $MgAl_2O_4(100)$ were partly replaced by Fe. $Ru_5O_{10}$ and $Ru_{10}O_{20}$ clusters that were cut from the $RuO_2$ crystal were employed as $RuO_2$ cluster models.

**Theoretical maximum loading of dispersed Ru atoms over spinel**. The BET surface area of Ru$_1$/MAFO-900 was 38 m$^2$ g$^{-1}$, hence 1 g of MAFO support provides 38 m$^2$ of surface ($S$) after 900 °C calcination. The spinels mainly have primary cuboctahedral shape with dominant {100} and {111} facets[44]. Assuming that all $M^{3+}$ on the surface can stabilize Ru atoms, the theoretical model indicates that the maximum density of atomically dispersed Ru ($D$) are 5.88 and 6.79 atom nm$^{-2}$ for {100} and {111} facets, respectively. The total number of isolated Ru atoms ($N$) that could be achieved for 1 g of Ru/MAFO is therefore predicted to be $N = D \times S$. Since

the mass of Ru equals $(N/N_A) \times M$, where $N_A$ is Avogadro's constant $(6.02 \times 10^{23}$ mol$^{-1}$), and $M$ is the molar mass of Ru (101 g mol$^{-1}$), the theoretical maximum loadings of isolated Ru atoms that could be dispersed over 1 g of MAFO are 3.7 and 4.3 wt% for {100} and {111} facets, respectively. Thus the calculated maximum Ru loading is about 4 wt% assuming that all $M^{3+}$ sites can stabilize Ru atoms. However, if only Fe$^{3+}$ can stabilize Ru, the maximum Ru loading should be 4 wt% $\times$ 0.8/2 = 1.6 wt% for MgAl$_{1.2}$Fe$_{0.8}$O$_4$ support. Similarly, the maximum Ru loading should be 4 wt% $\times$ 0.5/2 = 1.0 wt% for MgAl$_{1.5}$Fe$_{0.5}$O$_4$ support.

## Data availability

The data that support the findings of this study are available within the paper and its Supplementary Information, and all data are available from the authors on reasonable request.

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

## Acknowledgements

This paper is dedicated to the 70th anniversary of Dalian Institute of Chemical Physics, Chinese Academy of Sciences. This work was supported by National Key Projects for Fundamental Research and Development of China (2016YFA0202801), National Natural Science Foundation of China (21776270, 21972135, 21776271), Strategic Priority Research Program of the Chinese Academy of Sciences (XDB17020100), DNL Cooperation Fund, CAS (DNL180403), and LiaoNing Revitalization Talents Program (XLYC1807068). The synchrotron radiation experiment was performed at the BL14W1 at the Shanghai Synchrotron Radiation Facility, Shanghai Institute of Applied Physics, China.

## Author contributions

K.L. synthesized the catalysts and performed most of the reactions and collected and analyzed the data. X.Z. and T.Y. performed DFT calculations. G.R. analyzed part of the data. Y.R. and X.L. carried out the XAFS characterization. Y.S. and X.P. performed the scanning transmission electron microscopy characterization. J.Z., Z.C., and J.Y. provided reagents and performed some experiments. T. Zhou, W.X., and J.L. performed the aberration-corrected scanning transmission electron microscopy characterization. C.Z., H.M., W. Liu, and Q.J. performed the in situ electron microscopy. K.L., B.Q., A.F.L., and K.W. wrote the manuscript. B.Q., W. Li, A.W., and T. Zhang designed the study and supervised the project.

## Competing interests

The authors declare no competing interests.
