## [Peer Review File · Nature Communications]

Reviewers' comments:

Reviewer #1 (Remarks to the Author):

This manuscript reports the synthesis of Ru SACs by transformation of commercial RuO₂ powders on MgAl_{1.2}Fe_{0.8}O₄ spinel. The as-prepared Ru SACs exhibited superior nitrous oxide decomposition performance in comparison to RuO₂ aggregates. Significantly, the in situ characterization of RuO₂ dispersion was carried out to investigate the formation mechanism of Ru SACs. They found that RuO₂ powders undergo complete dispersion into SACs at 900 °C under oxidizing and inert atmospheres, and the strong metal-support interaction played the critical role for the transformation. Therefore, I recommend this manuscript to be accepted after solving the following issues.

1. The authors mentioned that Fe doping is very important. Can the Fe content be tuned in the spinel structure? Is there an optimized ratio for the formation of Ru SACs?
2. EXAFS fitting curves should be provided.
3. The authors claimed that the strong metal-support interaction between Ru and Fe plays the critical role for the formation of Ru atoms. The verification of the strong interaction should be further carried out. DFT was not enough.
4. In the DFT calculation, I suggest the authors could evaluate the interaction between MAFO and RuO₂. Once this is done, it can further confirm the stability observed by experimental side.
5. The loading amount of Ru is around 2%. Is it possible to increase the loading amount of Ru?
6. The Al, Mg and O signal should be shown in Figure S17.
7. The Ru single atoms in the supplementary Figures (such as Figure S3, S10, S12) should also be circled.
8. Some important references should be added: Energy Environ. Sci., 2019, 12, 492; Energy Environ. Sci., 2019, 12, 1000; Angew. Chem., Int. Ed., 2019, 58, 2622; Nat. Comm. 2017, 8, 1070; National Science Reviews 2018, 5, 628-630.

Reviewer #2 (Remarks to the Author):

Manuscript Review: NCOMMS-19-29451-T

Strong metal-support interaction promoted scalable production of thermally stable single-atom catalysts, by Liu, et al.

This paper describes the synthesis and characterization of a catalyst composed of Ru dispersed onto a Fe-doped MgAl_{1.2}Fe_{0.8}O₄ spinel support (although this level of Fe content, i.e., 27% of the metal content, goes beyond what I would consider “doping”). The authors show that the calcination temperature is crucial for dispersing the Ru from RuO₂ nanoparticles for T < 900 °C (using Ru acetylacetonate or RuO₂ precursors) to highly dispersed Ru for T ≥ 900 °C on the MAFO support. The XPS and EXAFS measurements, along with atomic compositions of the catalyst (e.g., ICP) make a fairly strong case for the dispersed state to be single Ru atoms, but I find the STEM images of the single atoms somewhat hard to see in Figure 1 (e, f), even with the circles; also, none of the STEM images in SI seem

to show single atoms (e.g., Figure S10 and S12). I realize it's a challenge to identify single atoms, but the author's claims of complete dispersion to single atoms is less important than the fact that the Ru cations are highly dispersed, especially compared to the RuOx particles at lower calcination temperatures.

A few specific comments:

1. The author's propose a Ostwald ripening mechanism to explain the Ru dispersion in which Ru atoms move from the larger RuOx particles to FeOx sites at the surface as a result of strong covalent metal-support interactions (CMSI). Firstly, XPS and XAS show that the atoms are Ru⁴⁺ cations and not neutral species so why couldn't they diffuse into the bulk (lattice substitution) rather than be just at the surface – the atoms have sufficient mobility at the high calcination temperatures used (≥ 900 °C) as evidenced by the loss of the larger RuO₂ nanoparticles. Clearly, FeOx is dispersed throughout the bulk and none of the techniques used, e.g., XAS/EXAFS, STEM, etc., can distinguish between Ru cations at the surface vs the bulk; XPS can be more surface sensitive, but not with a standard Al K α x-ray source (note the x-ray energy is incorrect on p. 15, line 387). The catalysis results suggest that the Ru cations are on the surface but that is not sufficient to argue that all the Ru atoms are at the surface.

2. Related to the above comments, I find the results of the DFT calculations to less than convincing (discussion bottom p. 10; SI, Figure 29). The authors argue that the Ru-O bonds at the interface with a Fe₂O₃ cluster (SI, Figure 29) is evidence for CMSI. I would that almost any oxide cluster could cause changes in the Ru-O bond length simply due to the opportunity for the metal cations at the interface to increase their O-coordination. This is not strong support for CMSI between Ru and FeOx, but it is clear from control experiments in which Ru dispersion is not observed for pure spinel (without Fe) and is observed for a pure Fe₂O₃ support that Ru has an affinity for Fe or FeOx. Perhaps there is less subtle reason for Ru(IV) incorporation into FeOx that has more to do with the RuOx-FeOx phase mixing that is driven by thermodynamics?

3. The author's list the Ru loading for each catalyst in Table S2, and I note that the Ru/Fe₂O₃ catalyst has the lowest loading of all (0.32%). Since the formation mechanism for single atoms is considered to be a result of strong interactions between FeOx and Ru, the authors should comment on why the Ru loading (0.31%) is so low on this sample, which represents their model for scale up.

Overall, I support publication of the manuscript after the author's consider the above comments.

Reviewer #3 (Remarks to the Author):

The authors have prepared a very nice paper on the scalable preparation of single atom Ru catalysts on spinels by use of strong metal support interactions to stabilize the catalytic sites. I found this to be both an important and well executed paper both from the technical as well as the presentation aspects. I have only a few suggestions for the authors to help improve the paper.

1, Does the scaled-up version of the catalysts described on pg 11 exhibit the same reactivity as the samples prepared on MAFO. My worry would be that the redox activity of the Fe_2O_3 may be different and also lead to other complexities under reaction conditions

2, Can the authors suggest other metals where this synthesis may or may not work based on their understanding of the mechanism.

Other than that, I enthusiastically support this paper for publication in Nature Comm.

Response to reviewers' comments and questions

Reviewer #1 (Remarks to the Author):

This manuscript reports the synthesis of Ru SACs by transformation of commercial RuO₂ powders on MgAl_{1.2}Fe_{0.8}O₄ spinel. The as-prepared Ru SACs exhibited superior nitrous oxide decomposition performance in comparison to RuO₂ aggregates. Significantly, the in situ characterization of RuO₂ dispersion was carried out to investigate the formation mechanism of Ru SACs. They found that RuO₂ powders undergo complete dispersion into SACs at 900 C under oxidizing and inert atmospheres, and the strong metal-support interaction played the critical role for the transformation. Therefore, I recommend this manuscript to be accepted after solving the following issues.

Response: Thank you for your positive comments.

Q1. *The authors mentioned that Fe doping is very important. Can the Fe content be tuned in the spinel structure? Is there an optimized ratio for the formation of Ru SACs?*

Response: The reviewer raised a very good question which is helpful to further clarify the role of Fe content in obtaining Ru SACs.

First of all, the Fe content can definitely be tuned in the spinel structure. Theoretically the Fe content in MgAl_{2-x}Fe_xO₄ spinel can be tuned from x = 0 to 2, i.e., from no substitution to total substitution of Al to form MgFe₂O₄ spinel. However, based on our experience, the doping of Fe will weaken the sintering resistance of the MgAl₂O₄ spinel thus arouses surface area decrease after being calcined at high temperatures. In addition, too much Fe doping will also result in the appearance of impure phase of iron oxide. Considering the critical role of Fe sites in stabilizing Ru atoms and the effect of Fe content on the spinel structure, we believe there's an optimized Fe ratio to get the exposed Fe sites maximized.

In order to experimentally demonstrate the above speculation, we prepared a few MgAl_{2-x}Fe_xO₄ (x = 0.5, 1, 1.5) spinel samples with different Fe contents by the same

preparation procedure. As shown in Figure R1a, both $\text{MgAl}_{1.5}\text{Fe}_{0.5}\text{O}_4$ and $\text{MgAl}_1\text{Fe}_1\text{O}_4$ samples showed pure crystalline spinel phase while $\text{MgAl}_{0.5}\text{Fe}_{1.5}\text{O}_4$ presented both spinel and iron oxide diffraction pattern, suggesting the upper limit of Fe doping should be lower than $x = 1.5$ for obtaining pure spinel structure. Meanwhile, with the Fe doping increasing, the surface area of spinels would gradually decrease (Table R1). We then tried to deposit Ru single atoms by using the newly synthesized spinels ($\text{MgAl}_{1.5}\text{Fe}_{0.5}\text{O}_4$ & $\text{MgAl}_1\text{Fe}_1\text{O}_4$) as supports with a nominal weight loading of 2 wt% with the same catalyst preparation procedure. As expected (Figure R1b), no RuO_2 peaks for the 2Ru/ $\text{MgAl}_1\text{Fe}_1\text{O}_4$ -900 sample were observed, suggesting the RuO_2 aggregates have been dispersed into single atoms, which was further confirmed by AC-HAADF-STEM characterization (Figure R2). However, weak diffraction peaks of RuO_2 were observed in the 2Ru/ $\text{MgAl}_{1.5}\text{Fe}_{0.5}\text{O}_4$ -900 sample, suggesting that RuO_2 cannot be completely dispersed on this support, probably due to the low Fe content in the $\text{MgAl}_{1.5}\text{Fe}_{0.5}\text{O}_4$ spinel which cannot provide sufficient sites to stabilize all Ru single atoms. In fact, the calculated theoretical maximum Ru loading for $\text{MgAl}_{1.5}\text{Fe}_{0.5}\text{O}_4$ is only about 1 wt% (see answer to Q5). Based on above analysis, we propose that for the catalyst with 2 wt% Ru loading the optimized Fe ratio should be around $x = 1$. But for lower Ru loading, the optimized Fe content needs further study and we think the principle is in the precondition of providing sufficient stabilizing sites, the smaller Fe content the better.

We have added these new results and discussion into our revised manuscript and SI.

Figure R1. XRD patterns of $\text{MgAl}_{2-x}\text{Fe}_x\text{O}_4$ ($x = 0.5, 0.8, 1, 1.5$) spinels (a) and

2Ru/MgAl_{2-x}Fe_xO₄-900 (x = 0.5, 0.8, 1) samples (b).

Figure R2. AC-HAADF-STEM images of 2Ru/MgAl₁Fe₁O₄-900 sample.

Table R1. Physicochemical properties of the spinels

Spinel	Surface area (m ² g ⁻¹)	Pore volume (cm ³ g ⁻¹)	Average pore size (nm)
MgAl _{1.5} Fe _{0.5} O ₄	132	0.36	8
MgAl _{1.2} Fe _{0.8} O ₄	109	0.34	10
MgAl ₁ Fe ₁ O ₄	99	0.37	13
MgAl _{0.5} Fe _{1.5} O ₄	36	0.15	15

Q2. EXAFS fitting curves should be provided.

Response: We thank the reviewer's suggestion. In the revised SI, we have provided the EXAFS fitting curves of Ru/MAFO samples (Supplementary Fig. 11 and 39b).

Q3. *The authors claimed that the strong metal-support interaction between Ru and Fe plays the critical role for the formation of Ru atoms. The verification of the strong interaction should be further carried out. DFT was not enough.*

Response: We thank the reviewer for the comments. First of all, we would like to clarify that the critical role of the strong interaction between Ru and FeO_x (CMSI) were verified not only by DFT calculation but also by a control experiment where on MgAl₂O₄ almost no dispersion of the RuO₂ aggregates was observed. In addition, the CMSI between Pt and FeO_x has been identified in our previous work (*Nat. Commun.* **2019**, 10, 234). In an unpublished work we have found the different interaction between Pt and MAFO with different calcination temperatures (see Figure R3). Therefore, we think the presence of CMSI between Ru and Fe is convincing. To further convince the reviewer, we performed H₂-TPR characterization of our Ru/MAFO & Ru/MA samples. As shown in Figure R4, for all non-SAC samples two reduction peaks were observed between 100 and 200 °C. The former corresponds to the reduction of RuO₂ to RuO while the latter can be ascribed to the reduction of RuO to Ru metal (*Catal. Commun.* **2007**, 8, 1531; *Cent. Eur. J. Chem.* **2013**, 11, 912). The slightly higher temperature for the reduction of RuO on Ru/MAFO-500 than that on Ru/MA-500 may suggest the Ru species interact stronger with MAFO than with MA support. Of more importance, the low-temperature reduction of Ru nearly vanished on the Ru₁/MAFO-900 sample with only a very tiny reduction peak (marked by arrow). The majority of the Ru species must have been reduced together with Fe at higher temperatures, suggesting a strengthened interaction between Fe and Ru after being calcined at 900 °C. A quantitative analysis (Table R2) revealed that the H₂ consumptions on all non-SACs are similar to the theoretical one for complete reduction of RuO₂ to Ru. However, the H₂ consumption of the tiny reduction peak is only about 1/27 of the theoretical one, corresponding to a reduction of Ru loading of ~0.07 wt%. We propose that these Ru species may be stabilized by Mg or Al sites since the MA support itself can stabilize very low loading of Ru single atoms (*ACS Catal.* **2019**, DOI: 10.1021/acscatal.9b03709).

Figure R3. H₂-TPR profiles of Pt/MAFO samples.

Figure R4. H₂-TPR profiles of Ru/MAFO and Ru/MA samples.

Table R2. H₂ consumptions of Ru/MAFO and Ru/MA samples based on H₂-TPR

Sample	H ₂ consumption ($\mu\text{mol/g}_{\text{cat}}$)	Corresponding amount of reduced Ru (wt %)
Ru/MA-500	437.1	2.21
Ru/MA-900	342.3	1.73
Ru/MAFO-500	356.3	1.80
Ru ₁ /MAFO-900	14.5	0.07

Considering that Ru existed in the form of Ru⁴⁺ and the weight loadings of samples were 2

wt%, thus the theoretical H₂ consumption for all samples is 395.8 μmol/g.

Q4. *In the DFT calculation, I suggest the authors could evaluate the interaction between MAFO and RuO₂. Once this is done, it can further confirm the stability observed by experimental side.*

Response: We thank the reviewer's good suggestion according to which we performed further DFT calculations to evaluate the interaction between RuO₂ and MAFO/MA (see Methods for calculation details or below). MA and MAFO were modeled by MgAl₂O₄(100) and two layer Fe-substituted MgAl₂O₄(100), respectively. RuO₂ was modulated by two clusters of Ru₅O₁₀ and Ru₁₀O₂₀. The relevant details are now included in the revised manuscript, SI, and below.

As shown in Figure R5 below, after optimization both clusters are obviously dispersed on Fe substituted MAFO support compared with on MA support. To further measure the dispersion magnitude, the distance between those four Ru atoms (#1, 3, 4, 5) and the Ru (#2) which could be viewed as the center of the cluster are summarized in Table R3. For both clusters either the longest or the average Ru–Ru distance increased obviously. In particular, on the basis of the longest or the average Ru–Ru distance values for the large Ru₁₀O₂₀ cluster in models c and d, the large cluster is easier to be dispersed by the substituted Fe. These results clearly revealed that the interaction between MAFO and RuO₂ comes from the substituted Fe atoms that are able to promote the dispersion of RuO₂ cluster.

Figure R5. Optimized structures for the Ru₅O₁₀ cluster on MgAl₂O₄(100) surface (a) and Fe-substituted MgAl₂O₄(100) surface (b) and the Ru₁₀O₂₀ cluster on MgAl₂O₄(100) surface (c) and Fe-substituted MgAl₂O₄(100) surface (d). In Fe-substituted MgAl₂O₄(100) model (b and d), Al atoms in top two layers of MgAl₂O₄(100) were partly substituted by Fe atoms. The Ru, O, Mg, Al, and Fe atoms are colored by blue, red, grey, pink, and green, respectively.

Table R3. The Ru–Ru(#2) distances (in Å) in different models in Figure R5.

Cluster	Ru ₅ O ₁₀			Ru ₁₀ O ₂₀		
	a	b	Increment	c	d	Increment
Ru(#1)–Ru(#2)	3.920	4.074		4.513	3.565	
Ru(#3)–Ru(#2)	3.305	3.620		4.207	4.941	
Ru(#4)–Ru(#2)	3.477	3.441		3.945	5.542	
Ru(#5)–Ru(#2)	3.556	3.610		4.252	3.542	
Average distance	3.565	3.686	3.4%	4.229	4.398	4.0%

Table R4. Binding energy E and Reaction energy G of the farthest RuO₂ dissociation at -273.15 °C, 800 °C, 900 °C, and 1000 °C.

Cluster	Ru ₅ O ₁₀		Ru ₁₀ O ₂₀	
Models	a	b	c	d
E (eV, -273.15 °C)	6.83	5.87	9.56	7.78
G (eV, -273.15 °C)	1.13	0.27	1.48	0.61
G (eV, 800 °C)	-1.89	-2.75	-1.54	-2.41
G (eV, 900 °C)	-2.23	-3.09	-1.88	-2.76
G (eV, 1000 °C)	-2.58	-3.44	-2.24	-3.11

To further confirm that the MAFO could promote the formation of separated RuO₂ molecule, the binding energy between the farthest RuO₂ moiety and rest part were calculated, as shown in Table R4. Moreover, the reaction Gibbs free energy (G) of the farthest RuO₂ dissociation was also calculated by using below chemical equations:

The entropy and enthalpy corrections to Gibbs free energy correction were calculated by taking into account the individual translational E_t and S_t , vibrational E_v and S_v , rotational E_r and S_r , and ZPE contributions. For slab models, the entropy and enthalpy corrections to free energies are neglected in this work. This method has been used in the previous study (*Nat. Commun.* **2019**, 10, 234).

As shown in Table R4, both the binding energy and reaction energy in models b and d are obviously lower than that in a and c, respectively, indicating that RuO₂ moiety dissociation occurs easily on Fe-substituted MgAl₂O₄(100) surface. As the temperature rises to 800 °C, the RuO₂ dissociation process has already become from endothermic to exothermic and the reaction Gibbs free energy in models b and d is still lower than that in a and c, respectively. These results revealed that RuO₂ dissociation from cluster on Fe-substituted MgAl₂O₄(100) surface is always thermodynamically preferred than that on

MgAl₂O₄(100) surface, suggesting that the MAFO surface could facilitate the formation of single Ru atom.

Figure R6. Electron density difference maps (isovalue = 0.009) for the Ru₅O₁₀ cluster on MgAl₂O₄(100) surface (a) and Fe-substituted MgAl₂O₄(100) surface (b) and the Ru₁₀O₂₀ cluster on MgAl₂O₄(100) surface (c) and Fe-substituted MgAl₂O₄(100) surface (d). The charge depletion and accumulation regions are shown in blue and yellow, respectively.

The electron density difference, which is defined by the difference between the electron density of the RuO₂-cluster/surface system and the sum of the electron density of the deformed surface and a deformed isolated RuO₂ cluster, have been calculated, as shown in Figure R6. By comparing the models a (or c) and b (or d), it is clear that even the Fe in the second layer of models b (or d) also participate the charge transfer between the surface and cluster, revealing the Fe substitution effect on the interaction between the surface and cluster.

Computational methods

All DFT calculations were performed with Vienna Ab-initio Simulation Package (VASP) and the exchange-correlation energy was expressed by generalized gradient approximation (GGA) of Perdew-Burke-Ernzerhof (PBE) functional. The projector-augmented wave (PAW) method was used to describe the interaction between electrons and ions. The plane-wave basis energy cutoff was set to 520 eV with the gamma point only for the Brillouin zone. The

convergence criteria for the electronic structure and geometry optimization were 1×10^{-4} eV and 0.02 eV/Å, respectively. Because of the strongly correlated d electrons, DFT + U calculations with corresponding U-J values of 2.5 eV (Fe) and 2.0 eV (Ru) were employed.

Computational models

The 2×2 supercell model of $\text{MgAl}_2\text{O}_4(100)$ consists of four Al-O layers and three Mg layers, of which bottom two layers were fixed in the relaxation calculations. A 15 Å vacuum layer was added to avoid interaction between periodic structures. To model the MAFO, Al in top layers of $\text{MgAl}_2\text{O}_4(100)$ were partly replaced by Fe, Ru_5O_{10} and $\text{Ru}_{10}\text{O}_{20}$ clusters which were cut from the RuO_2 crystal were employed as RuO_2 cluster models.

Q5. *The loading amount of Ru is around 2%. Is it possible to increase the loading amount of Ru?*

Response: The reviewer raised a very good question. This question is actually somewhat linked to **Q1**. To answer this question, we have prepared Ru/MAFO samples with higher Ru nominal weight loadings of 2.5 & 3.0 wt%. Unfortunately, obvious RuO_2 diffraction peaks were observed in both samples (Figure R7), indicating that it is difficult to synthesize SACs with Ru loading higher than 2 wt%.

We then calculated the theoretical maximum loading of dispersed Ru atoms over MAFO support by using the method of our previously reported paper (*Nat. Commun.* **2019**, 10, 234, details see below and the revised SI). The calculated maximum Ru loading is about 4 wt% assuming all M^{3+} sites can stabilize Ru atoms. However, if only Fe^{3+} can stabilize Ru, the maximum Ru loading should be $4 \text{ wt\%} \times 0.8 / 2 = 1.6 \text{ wt\%}$, in good consistent with our experimental result of upper limit of $\sim 2 \text{ wt\%}$.

Calculation details:

The BET surface area of $\text{Ru}_1/\text{MAFO-900}$ was $38 \text{ m}^2 \text{ g}^{-1}$, hence 1 g of MAFO support provides 38 m^2 of surface (S) after 900 °C calcination. The spinels mainly have primary cuboctahedral shape with dominant {100} and {111} facets (*Nat. Commun.* **2013**, 4, 2481).

Assume that all M^{3+} on the surface can stabilize Ru atoms. The theoretical model indicates that the maximum density of atomically dispersed Ru (D) are 5.88 and 6.79 atom nm^{-2} for {100} and {111} facets, respectively. The total number of isolated Ru atoms (N) that could be achieved for 1 g of Ru/MAFO is therefore predicted to be $N = D \times S$. Since the mass of Ru equals $(N / N_A) \times M$, where N_A is Avogadro's constant ($6.02 \times 10^{23} \text{ mol}^{-1}$), and M is the molar mass of Ru (101 g mol^{-1}), the theoretical maximum loadings of isolated Ru atoms that could be dispersed over 1 g of MAFO are 3.7 and 4.3 wt% for {100} and {111} facets, respectively.

Figure R7. XRD patterns of 2.5/3Ru/MAFO-900 samples.

Q6. The Al, Mg and O signal should be shown in Figure S17.

Response: We thank the reviewer's good suggestion. We have shown these signals in the revision (Supplementary Fig. 21).

Q7. The Ru single atoms in the supplementary Figures (such as Figure S3, S10, S12) should also be circled.

Response: We thank the reviewer's good suggestion. We have circled the Ru single atoms in SI.

Q8. *Some important references should be added: Energy Environ. Sci., 2019, 12, 492; Energy Environ. Sci., 2019, 12, 1000; Angew. Chem., Int. Ed., 2019, 58, 2622; Nat. Comm. 2017, 8, 1070; National Science Reviews 2018, 5, 628-630.*

Response: Thank you for kindly bringing these nice references. We have cited them in our revision.

Reviewer #2 (Remarks to the Author):

This paper describes the synthesis and characterization of a catalyst composed of Ru dispersed onto a Fe-doped $\text{MgAl}_{1.2}\text{Fe}_{0.8}\text{O}_4$ spinel support (although this level of Fe content, i.e., 27% of the metal content, goes beyond what I would consider “doping”). The authors show that the calcination temperature is crucial for dispersing the Ru from RuO_2 nanoparticles for $T < 900$ °C (using Ru acetylacetonate or RuO_2 precursors) to highly dispersed Ru for $T 900$ °C on the MAFO support. The XPS and EXAFS measurements, along with atomic compositions of the catalyst (e.g., ICP) make a fairly strong case for the dispersed state to be single Ru atoms, but I find the STEM images of the single atoms somewhat hard to see in Figure 1 (e, f), even with the circles; also, none of the STEM images in SI seem to show single atoms (e.g., Figure S10 and S12). I realize it’s a challenge to identify single atoms, but the author’s claims of complete dispersion to single atoms is less important than the fact that the Ru cations are highly dispersed, especially compared to the RuO_x particles at lower calcination temperatures.

Response: Thank you for your nice comments and good suggestions which are very helpful in improving our manuscript. The Fe content in our $\text{MgAl}_{1.2}\text{Fe}_{0.8}\text{O}_4$ sample is indeed quite high. But for spinel materials the term of doping is often used (*Chem. Rev.* **2017**, 117, 10121) and according to the definition therein (the cation-doped spinel oxides can be expressed as $\text{A}_{1-x}\text{A}'_x\text{B}_{2-y}\text{B}'_y\text{O}_4$ ($0 \leq x/y \leq 1$)) our Fe modified spinel can be called as “doping”. However, we have the same feeling with you that using doping is somewhat misleading, so in the revision we have changed the term “doped” into “substituted” which will not result in ambiguity anymore.

We acknowledge that the contrast between Ru and MAFO support is relatively low in AC-HAADF-STEM images, which is due to the small difference of atomic number between Ru and support elements such as Al, Fe and Mg. The contrast of this type of images, the so called Z contrast, comes majorly from the difference in atomic number of the examined elements. Nevertheless, even though the contrast is relatively low, isolated Ru single atoms could be still identified. Of more importance, if there were small NPs/clusters on the support,

they could be easily observed and discriminated (see Supplementary Fig. 21, small RuO₂ NPs or nanoclusters on MA support).

A few specific comments:

Q1. *The author's propose an Ostwald ripening mechanism to explain the Ru dispersion in which Ru atoms move from the larger RuO_x particles to FeO_x sites at the surface as a result of strong covalent metal-support interactions (CMSI). Firstly, XPS and XAS show that the atoms are Ru⁴⁺ cations and not neutral species so why couldn't they diffuse into the bulk (lattice substitution) rather than be just at the surface – the atoms have sufficient mobility at the high calcination temperatures used (900 °C) as evidenced by the loss of the larger RuO₂ nanoparticles. Clearly, FeO_x is dispersed throughout the bulk and none of the techniques used, e.g., XAS/EXAFS, STEM, etc., can distinguish between Ru cations at the surface vs the bulk; XPS can be more surface sensitive, but not with a standard Al K α x-ray source (note the x-ray energy is incorrect on p. 15, line 387). The catalysis results suggest that the Ru cations are on the surface but that is not sufficient to argue that all the Ru atoms are at the surface.*

Response: The reviewer raised a very good question. Firstly, we would like to thank the reviewer for kindly pointing our mistake. We have corrected the X-ray energy to 1486.6 eV. Second, we acknowledge that the Fe is dispersed throughout the bulk and none of the techniques used in this work can sensitively distinguish the position of Ru on surface or not. However, both of our previous work (*Nat. Commun.* **2019**, 10, 234) and Prof. Dayte's work (*Science* **2016**, 353, 150) demonstrated that, based on Ion Scattering Spectroscopy (ISS) and other measurement, the single atoms would not diffuse into the bulk of the support. According to these work, and together with the fact that our Ru SAC in this work can have a maximum loading of only 2 wt% (see response to **Q5** of reviewer #1), we can conclude that the majority of Ru existed on the support surface or sub-surface because if they can diffuse into bulk then the maximum loading should be much higher. However, we don't have a certain explanation why Ru atoms would not diffuse into the bulk. Our thought is that Ru

atoms existed in Ru^{4+} , when they diffused into the bulk, they should either occupy the position of Fe^{3+} and Al^{3+} to form ion vacancy or locate at the interstitial position of the crystal to form interstitial defect. Defects are relatively less stable at high temperatures thus the diffusion of Ru into bulk may be thermodynamically unfavorable.

Q2. Related to the above comments, I find the results of the DFT calculations to less than convincing (discussion bottom p. 10; SI, Figure 29). The authors argue that the Ru-O bonds at the interface with a Fe_2O_3 cluster (SI, Figure 29) is evidence for CMSI. I would that almost any oxide cluster could cause changes in the Ru-O bond length simply due to the opportunity for the metal cations at the interface to increase their O-coordination. This is not strong support for CMSI between Ru and FeO_x , but it is clear from control experiments in which Ru dispersion is not observed for pure spinel (without Fe) and is observed for a pure Fe_2O_3 support that Ru has an affinity for Fe or FeO_x . Perhaps there is less subtle reason for Ru(IV) incorporation into FeO_x that has more to do with the RuO_x - FeO_x phase mixing that is driven by thermodynamics?

Response: The reviewer raises a very good question.

1) We totally agree with you that maybe any oxide clusters could cause changes in the Ru-O bond length. But we think the degree of this change may be different. To verify this, we have calculated the Al_2O_3 cluster onto the RuO_2 surface for comparison. As shown in Figure R8 and Table R5, the presence of the Al_2O_3 cluster on the RuO_2 surface causes smaller changes in Ru-O bond length, compared to the elongated Ru-O bond lengths caused by the Fe_2O_3 cluster, reflecting a stronger interaction between RuO_2 and FeO_x . But considering that we have demonstrated the role of strong interaction between FeO_x and RuO_2 by DFT calculation according to reviewer #1's suggestion (see answer to **Q4** of reviewer #1), we would like to remove the DFT data of this part in our revision. 2) Theoretically, RuO_x - FeO_x phase mixing could be another driven force for the dispersion of RuO_2 if they are totally mixable. However, according to our experimental data that the maximum loading of Ru on Fe_2O_3 is quite low (only 0.3 wt%), we think this is not the case.

Figure R8. Initial structures of Fe_8O_{12} (a) and Al_8O_{12} (b) clusters and optimized structures of Fe_8O_{12} (c) and Al_8O_{12} (d) on the $\text{RuO}_2(110)$ surface. The initial structures of Fe_8O_{12} and Al_8O_{12} clusters were cut from the $\alpha\text{-Fe}_2\text{O}_3(001)$ and $\alpha\text{-Al}_2\text{O}_3(001)$ surfaces, respectively. The Ru, O, Fe, and Al atoms are colored by blue, red, grey, and pink, respectively.

Table R5. The Ru–O lengths (in Å) in different models in Figure R8

Models	$\text{Fe}_8\text{O}_{12}/\text{RuO}_2(110)$	$\text{Al}_8\text{O}_{12}/\text{RuO}_2(110)$
1	2.019	2.047
2	2.003	1.966
3	1.993	1.964
4	1.980	1.955
5	1.977	1.945
6	1.960	1.936
7	1.896	1.848
Average length	1.975	1.952

Q3. *The author's list the Ru loading for each catalyst in Table S2, and I note that the Ru/Fe₂O₃ catalyst has the lowest loading of all (0.32%). Since the formation mechanism for single atoms is considered to be a result of strong interactions between FeO_x and Ru, the authors should comment on why the Ru loading (0.31%) is so low on this sample, which represents their model for scale up.*

Response: The reviewer raised a good question. As discussed above, for Ru SACs, Ru atoms were mainly located at the surface or subsurface of the support. The surface area of Fe₂O₃ after calcination at 900 °C is only ~4 m² g⁻¹ which can stabilize a maximum Ru loading of ~0.4 wt%. Hence when we tried to scale up the synthesis we used a low Ru loading of 0.3 wt% to ensure that the support can provide sufficient stabilization sites. Actually the property of high surface area and better sintering resistance is one of the advantages of MAFO used as a catalyst support. The only reason we use Fe₂O₃ rather than MAFO as a support to perform the scale-up preparation is that there's no commercially available MAFO support yet.

Overall, I support publication of the manuscript after the author's consider the above comments.

Response: Thank you for your support.

Reviewer #3 (Remarks to the Author):

The authors have prepared a very nice paper on the scalable preparation of single atom Ru catalysts on spinels by use of strong metal support interactions to stabilize the catalytic sites. I found this to be both an important and well executed paper both from the technical as well as the presentation aspects. I have only a few suggestions for the authors to help improve the paper.

Response: We thank this reviewer for the positive comments.

***Q1.** Does the scaled-up version of the catalysts described on pg 11 exhibit the same reactivity as the samples prepared on MAFO. My worry would be that the redox activity of the Fe₂O₃ may be different and also lead to other complexities under reaction conditions.*

Response: This is a very good question and similar to **Q3** of reviewer #2. We totally understand your concern and we also believe the Ru/Fe₂O₃ would have lower activity due not only to their different redox activity but also of the significantly lower surface area. The reason we used Fe₂O₃ rather than MAFO as a support to do the scale-up synthesis is that MAFO is commercially unavailable. We have tested the activity of the Ru₁/Fe₂O₃-1000g-900 catalyst in low concentration N₂O decomposition reaction under the premise of using same Ru amount (reaction conditions: 670 mg catalyst; gas flow, 33.3 mL min⁻¹). As expected, the activity of the Ru₁/Fe₂O₃-1000g-900 is indeed much lower than that of Ru₁/MAFO-900 catalyst, Figure R9.

Figure R9. N₂O conversion as a function of reaction temperature over Ru₁/MAFO-900 and Ru₁/Fe₂O₃-1000g-900 SACs at low concentration (1000 ppm N₂O, Ar balance). Reaction conditions: 100 mg Ru₁/MAFO-900 catalyst or 670 mg Ru₁/Fe₂O₃-1000g-900 catalyst; gas flow, 33.3 mL min⁻¹.

Q2. Can the authors suggest other metals where this synthesis may or may not work based on their understanding of the mechanism.

Response: This is a very good question. Actually we had performed some primary screening of different noble metals and found that Rh and Ir SACs can be synthesized by using this method. As shown in Figure R10, only spinel diffraction peaks were exhibited in both Rh and Ir SACs and AC-HAADF-STEM and EDX elemental mapping images confirmed the atomically dispersed Rh and Ir single atoms, Figure R11-12. However, for Pt and Pd metals SACs cannot be obtained by using PdO and PtO₂ as precursors. The reasons are not clearly yet at this stage and further study is in progress and results may be published independently later on.

Figure R10. XRD patterns of the 1Rh/MAFO-900 and 1Ir/MAFO-900 samples.

Figure R11. AC-HAADF-STEM images of 1Rh/MAFO-900 sample (a) and (g-h) and corresponding energy dispersive X-ray spectroscopy element mapping of Mg (b), Al (c), Fe (d), O (e) and Rh (f) of image (a).

Figure R12. AC-HAADF-STEM images of 1Ir/MAFO-900 sample (a) and (g-h) and corresponding energy dispersive X-ray spectroscopy element mapping of Mg (b), Al (c), Fe (d), O (e) and Ir (f) of image (a).

REVIEWERS' COMMENTS:

Reviewer #1 (Remarks to the Author):

I think that the authors have well addressed all comments and suggestions with some additional new data. I think that it is publishable.

Reviewer #2 (Remarks to the Author):

the author's responses to my comments and those of the other reviewers is satisfactory and have improved the manuscript by strengthening arguments for the formation of Ru single atoms on the Fe substituted spinel support. I am still skeptical as to the value of the DFT calculations, but they are not crucial for supporting the main conclusions of the paper regarding mechanism of RuO₂ decomposition and dispersion at high temperatures. I therefore support publication of the revised manuscript.

Reviewer #3 (Remarks to the Author):

I am pleased with the care and detail of the authors response to my, and the other reviewers, questions. I believe the paper is suitable for Nat. Comm.